# Examining Relationships between Heat Requirement of Remotely Sensed Green-up Date and Meteorological Indicators in the Hulun Buir Grassland

**Jian Guo [1,2], Xiuchun Yang [1,\*], Fan Chen [3], Jianming Niu [2], Sha Luo [4], Min Zhang [5], Yunxiang Jin [6], Ge Shen [7], Ang Chen [1], Xiaoyu Xing [6], Dong Yang [6] and Bin Xu [6]**

[1] School of Grassland Science, Beijing Forestry University, Beijing 100083, China; guojian@mail.imu.edu.cn (J.G.); chenang0226@163.com (A.C.)
[2] School of Ecology and Environment, Inner Mongolia University, Hohhot 010021, China; jmniu2005@163.com (J.N.)
[3] Academy of Agricultural Planning and Engineering, Ministry of Agriculture and Rural Affairs, Beijing 100125, China; iarrp0720@163.com (F.C.)
[4] Department of Electrical and Computer Engineering, National University of Singapore, Singapore 117583, Singapore; eleluos@nus.edu.sg (S.L.)
[5] Faculty of Geoscience and Environmental Engineering, Southwest JiaoTong University, Chengdu 611756, China; 3541190392@163.com (M.Z.)
[6] Key Laboratory of Agri-informatics, Ministry of Agriculture and Rural Affairs/Institute of Agricultural Resources and Regional Planning, Chinese Academy of Agricultural Sciences, Beijing 100081, China; jinyunxiang@caas.cn (Y.J.); 82101185223@caas.cn (X.X.); yd0696@163.com (D.Y.); xubin@caas.cn (B.X.)
[7] Key Laboratory of Agricultural Remote Sensing (AGRIRS), Ministry of Agriculture and Rural Affairs/Institute of Agricultural Resources and Regional Planning, Chinese Academy of Agricultural Sciences, Beijing 100081, China; 82101181153@caas.cn (G.S.)
\* Correspondence: yangxiuchun@bjfu.edu.cn

**Abstract:** The accumulation of heat and moderate precipitation are the primary factors that are used by grasslands to trigger a green-up date. The accumulated growing degree-days (AGDD) requirement over the preseason is an important indicator of the response of grassland spring phenology to climate change. This study adopted the Normalized Difference Phenology Index (NDPI), which derived from the Moderate Resolution Imaging Spectroradiometer (MODIS), to extract annual green-up dates in the Hulun Buir grassland in China between 2001–2015. Our analysis indicated that the range (standard deviation) and trend for the green-up date were DOY (day of year) 104 to DOY 144 (10.6 days) and -2.0 days per decade. Nine point two percent of the study area had significant ($p < 0.05$) changes in AGDD requirements. The partial correlations between the AGDD requirements and chilling days (67.04%, pixels proportion) were negative and significant ($p < 0.05$). The partial correlations between the AGDD requirement and precipitation (28.87%) were positive and significant ($p < 0.05$). Finally, the partial correlation between the AGDD requirement and insolation (97.65%) were positive and significant ($p < 0.05$). The results of this study could reveal the response of vegetation to climate warming and contribute to improving the phenological mechanism model of different grassland types in future research.

**Keywords:** green-up date; accumulated growing degree-days; climate change; meteorological indicators; Hulun Buir; grassland

## 1. Introduction

Plant phenology aims to investigate the interrelationships between cyclical plant phenological phenomena and environmental conditions [1,2]. Phenological events reflect a combination of intrinsic factors of plants (genetic) and environmental impacts. Since the avoidance of late spring frost damage and early autumn cold events are very important

for the survival of plants, timely green-up is essential [3]. Changes in the phenological phenomena of vegetation that are caused by global climate changes lead to constant changes in a variety of ecosystem processes, which, in turn, could affect weather and climate systems [4–6]. Recent studies have increasingly emphasized the impact of climate change on phenology [7–11]. Plants in temperate and frigid regions require an adequate amount of heat with which to initiate green growth in the spring or early summer [12,13]. This specific amount of heat is also referred to as the heat requirement, which is often calculated using the classical growing degree-days concept [14–16]. A parameter referred to as accumulated growing degree-days (AGDD) has been widely utilized as a critical parameter in process-based phenological models in order to simulate plant growth [1,17,18]. In grasslands, the green-up date reflects the important turning point of the process in which the grassland begins to germinate, turn green, and grow; these processes happen when temperature and water conditions reach a suitable state in the spring. Vegetation green-up date characterizes the onset of photosynthesis on land and can serve as a good indicator to measure terrestrial ecosystem processes such as carbon and water cycles and the energy balance between the biosphere and the atmosphere [19–21]. Moreover, the vegetation green-up date has been identified as one of the simplest and most sensitive indicators in measuring the response of grassland ecosystems to climate change, particularly in high latitudes and high altitudes [22].

Some phenological models, based on the growing degree-days, have been widely used to simulate plant phenology and receive much attention [23,24]. In these phenological models, a critical threshold for AGDD requirement is species-specific and location-specific; then, these models can determine the green-up date [17,25]. The thresholds of AGDD requirement were determined commonly by experiments or optimization processes in process-based phenology models. The interaction of heat requirement with contextual environmental factors could help increase our understanding of the global carbon balance in climate warming and improve spring phenological predictions [18].

In recent decades, many studies have examined the AGDD of woody plants [12,26,27]. For example, among 13 temperate species in Europe and North America, the AGDD requirement for leaf unfolding increased by 50% between 1980 and 2012 [27]. In addition, the AGDD requirement showed considerable spatial variation for vegetation green-up in north-central and high latitudes [28,29]. This indicated that temperature might not be the only factor influencing spring phenology. Therefore, it is necessary to put other meteorological indicators in the model to enhance the fitting accuracy. It is very important that we improve our understanding of the AGDD requirements for spring phenology and its spatiotemporal changes. Early field observations recorded the AGDD requirement of several woody plants in Europe and North America; the analysis showed that temperate late-spring species had a greater heat requirement than early spring species [12]. In the temperate steppe and forests in the Northern Hemisphere, the AGDD requirements showed extensive spatial positive correlations with vegetation green-up date and preseason precipitation [28]. Several process-based models combining AGDD and precipitation for fitting grassland green-up onset dates have been developed in Inner Mongolia [30,31] and the Qinghai-Tibet Plateau [1,32], where the correlation coefficient for simulating interannual variations of green-up date was low [1]. Recent research has improved our understanding of the regulatory effects of the photoperiod on the heat requirements of woody plants [27,33]. In the Qinghai-Tibet Plateau region of China, temperatures increased rapidly between 1998 and 2012, but no significant increase in AGDD requirements was observed [34].

In Chinese grassland areas, ground observation records and satellite data have been increasingly used to identify temporal changes in the green-up date and the relationship between these changes and climate [35]. Most of these studies have focused on the Qinghai-Tibet Plateau [34,36–39]. The general belief was that vegetation required a certain amount of heat accumulation to trigger green-up in the spring [37,40]. Moreover, the changes in vegetation phenology in the spring may exert influences on the biophysical

processes on the land surface and may even affect the regional climates in East Asia [41]. The effects of environmental factors on vegetation green-up have also been explored, such as winter temperatures and their relationship with cold effects [38], preseason precipitation [42], photoperiod, and sunshine hours [43]. Theoretically, even if the temperature during the vegetation dormant period differed annually, the heat requirement for plant green-up remained relatively constant. A previous study reported that the heat requirement for vegetation green-up on the Qinghai-Tibet Plateau did not change significantly although the AGDD requirements showed a negative correlation with chilling days (CD) [34]. Generally, enough chilling days were needed to break dormancy before the green-up date [44].

Previous research aimed to investigate how the interannual variability of dormant CD affects the heat requirements at the species and regional scales [27,34]. However, only a limited number of studies have focused on different types of grasslands. In addition to the accumulation of CD during dormancy, the heat requirements of grassland vegetations may also be affected by other environmental factors, such as precipitation and insolation. The analysis of the temporal and spatial variations in the heat requirements of spring vegetation in grasslands and their interactions with environmental factors could improve our understanding on the impact of meteorological indicators before the green-up date. [19,27,34,43].

Our present research focuses on the Hulun Buir grassland for two main reasons. First, this grassland is distributed across the extreme northeast of China. Second, Hulun Buir is the most concentrated and representative area of a temperate meadow steppe in China. Our study has three main aims: (1) to monitor the vegetation green-up dates between 2000 and 2015 and analyze the temporal and spatial patterns by the mean (± standard deviation) and the trends for variations in the green-up dates; (2) to analyze the trends in variations of the AGDD requirements and other meteorological indicators (CD, precipitation, and insolation); and (3) to investigate the impact of meteorological indicators on the interannual variations in the AGDD requirement in different types of grasslands.

## 2. Materials and Methods

### 2.1. Study Area

The Hulun Buir grassland occupies the highest latitude of Northern China. The temperature in this region increased significantly between 1970 and 2014, with the mean temperature increasing by 0.2°C per decade [45]. In arid regions, evaporation was significantly higher than precipitation, and severe droughts often occurred in the spring and early summer [46]. The average annual temperature range of the Hulun Buir grassland was -8.6–4.4°C during 1979-2015 (Figure 1), and the average annual precipitation was 255–588 mm (Figure 2), gradually decreasing from east to west. Another characteristic of the precipitation in this area is the uneven temporal distribution, which includes annual and seasonal variations, with most of the precipitation occurring in July and August.

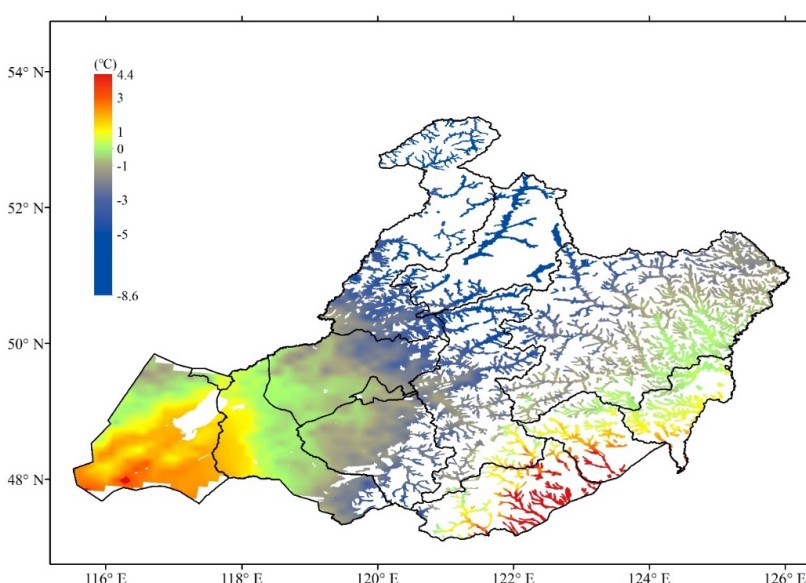

**Figure 1.** The multiyear annual average temperature during 1979-2015.

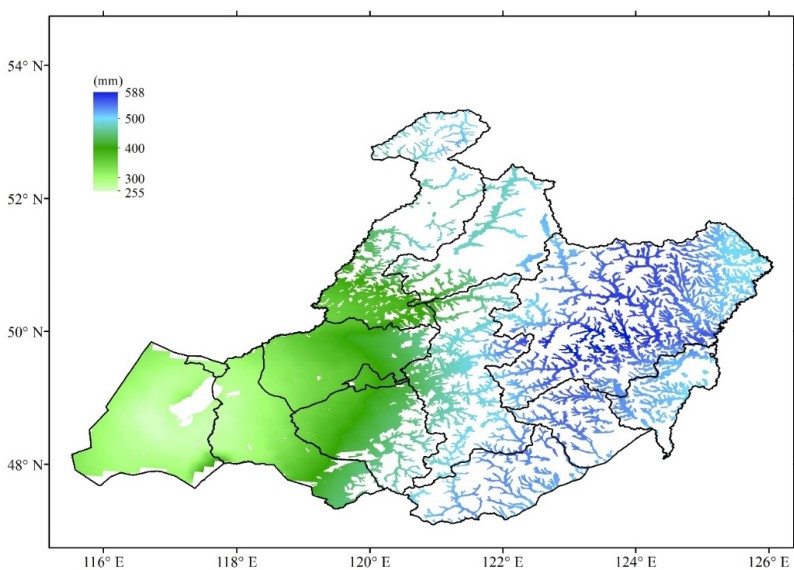

**Figure 2.** The multiyear annual average precipitation during 1979-2015.

The Hulun Buir grassland covers an area of approximately 100,000 km² in the middle latitudes of Eurasia, between 115°31′E and 126°04′E and 47°05′N and 53°20′N. The region is located northeast of the temperate grasslands in China. Due to the high latitude, the heat from the ground radiation is reduced, and the temperatures are relatively low. This grassland lies far from the direct influence of the ocean. In addition, this area is controlled by the Mongolian high-pressure air mass. Therefore, the Hulun Buir grassland has a temperate continental monsoon climate. The Greater Khingan Mountains run northeast to southwest and through the center of Hulun Buir City, thus reducing the chances of monsoons from the Southeast Pacific reaching deep into the mainland. Simultaneously, because of the natural barrier created by the Greater Khingan Mountains, the cold current from Siberian Mongolia is blocked. Thus, there is a clear difference in climates on the two sides of these mountains. To the east of the Greater Khingan Mountains, there is a semi-humid forest grassland climate with four distinct seasons, a mild climate, and heavy rainfall. In comparison, the west side of the Greater Khingan Mountains has a semi-humid and semi-arid grassland climate and is colder and drier with less rainfall. The Greater Khingan Mountains form a cold and humid forest climate [47].

Hulun Buir is the most concentrated and representative area of a temperate meadow steppe in China, in which many types of meadow steppe ecosystems develop. From the east humid region to the west semi-arid region, an ecological geographical gradient of climatic dryness is formed from east to west (Figure 3). Thus, zonal vegetation is divided from east to west, with a temperate meadow steppe and temperate steppe predominant in the west, lowland meadow, and upland meadow in the east [48].

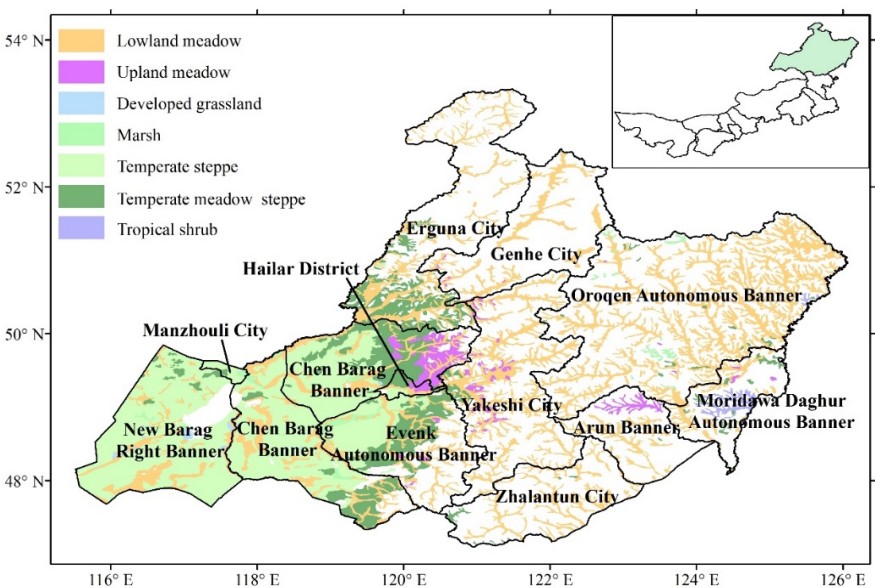

**Figure 3.** Administrative divisions and grassland classification in Hulun Buir. In this figure, county boundaries were used to divide the different districts. The name of each administrative district is given in bold. The blank areas represent non-grassland areas. The grassland classification dataset is based on a 1:1,000,000 Vegetation Atlas of China [49].

### 2.2. Datasets

We downloaded Moderate Resolution Imaging Spectroradiometer (MODIS) MOD09A1 data (2001-2015) National Aeronautics and Space Administration (NASA) https://modis.gsfc.nasa.gov/. MOD09A1 provides 8-day surface reflectance of the MODIS Terra product with 500-meter resolution of bands 1-7, projected as sinusoidal projection. Each MOD09A1 pixel contains the best possible L2G observations over an 8-day period, taking into account the effects of high viewing coverage, low viewing angles, cloudless and cloudless shadows, and aerosol concentrations. This scientific dataset provides products, including reflectance values for bands 1-7 and quality evaluation data. Data project transformation and subset for analysis were prepared using the MODIS Reprojection Tool and Envi5.3 software. For processing, the time series was filtered with a double logistic function. The function formula used was given below:

$$g(t; x_1, \dots, x_4) = \frac{1}{1 + \exp\left(\frac{x_1 - t}{x_2}\right)} - \frac{1}{1 + \exp\left(\frac{x_3 - t}{x_4}\right)} \tag{1}$$

where $x_1$ determines the position of the Normalized Difference Phenology Index (NDPI) time series curve rising inflection point, while $x_2$ gives the rate of change. Similarly, $x_3$ determines the position of the NDPI time series curve falling inflection point, while $x_4$ gives the rate of change at this point. Additionally, for this function, the parameters are restricted in range to ensure a smooth shape.

The meteorological data used in this study included temperature and precipitation data downloaded from the China Meteorological Forcing Dataset [50], with a spatial resolution of 0.1° and a temporal resolution of 3 h. Meteorological data were resampled from 0.1° to 500 m using the Nearest-Neighbors method in Envi5.3 software.

### 2.3. Determination of Vegetation Green-Up Date

Remote sensing data have been widely used in the large-scale monitoring of vegetation activities, particularly for grassland ecosystems [51–53]. The green-up date determined by remote sensing reflects vegetation at a regional scale and is based on a time series of vegetation index. Commonly used vegetation indices include the normalized difference vegetation index (NDVI) and the enhanced vegetation index (EVI). When calculated by NDVI, green-up date has been shown to be affected by snowmelt in boreal regions [54]. The normalized difference phenology index (NDPI) was therefore developed to minimize the effects of snowmelt [55]. Commonly, three methods of extraction are used for green-up dates: dynamic threshold, maximum slope, and median methods. A comparative analysis showed that the median method is the most reliable [56].

Due to the significant influence of snowmelt in the Hulun Buir grassland, we chose to use the NDPI in this paper for phenological period extraction. The contrasting results between the NDVI and NDPI at pixels scaling is shown in Figure 4. The NDVI before growing season had many abnormal values. Thus, NDPI could better reflect the growth of vegetation. The median method was used to monitor the annual green-up date. The formula used was given below:

$$NDPI_{mid} = (NDPI_{max} + NDPI_{min}) \times 50\%, \tag{2}$$

where $NDPI_{mid}$ represents the NDPI of the green-up date, $NDPI_{max}$ represents the maximum NDPI throughout the growing season, and $NDPI_{min}$ represents the minimum of the NDPI increase phase.

**(a)**

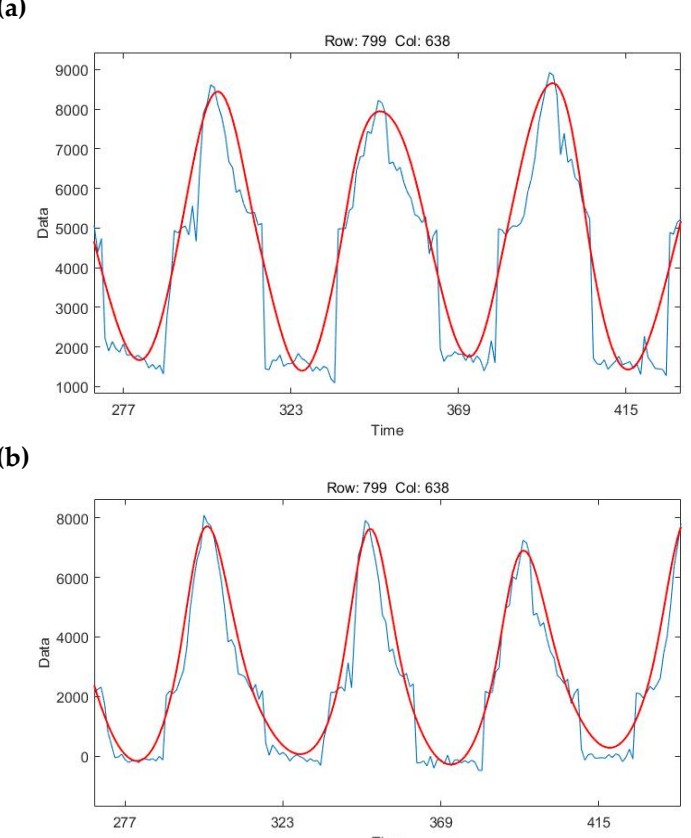

**(b)**

**Figure 4.** Time series reconstruction results under the double-logistic base normalized difference phenology index (NDPI) (**a**) and normalized difference vegetation index (NDVI) (**b**).

### 2.4. Calculation on the Key Heat Requirement and Meteorological Indicators

The AGDD requirement was chosen for heat requirement, and number of CD, precipitation, and insolation were chosen for the key meteorological indicators. The annual AGDD requirement for the vegetation green-up was defined as the sum of daily mean temperatures above 0°C in the preseason (1st January to green-up date) [12,57].

$$AGDD = \sum_{t_0}^{t_{GD}} T_t \qquad if \ T_t > T_0,$$ (3)

where AGDD represents the effective accumulated temperature for the vegetation green-up date, $t_0$ represents 1st January, $t_{GD}$ represents the vegetation green-up date, and $T_t$ represents the daily mean temperature.

CD was defined as the number of days over the same period with mean daily temperatures below 0°C [34].

$$CD = \sum_{t_0}^{t_{GD}} D_t \qquad D_t = \begin{cases} 1, & T_t < 0 \\ 0, & T_t \geq 0' \end{cases}$$ (4)

where CD represents the number of chilling days for vegetation green-up date, $t_0$ represents 1st January, $t_{GD}$ represents the vegetation green-up date, and $T_t$ represents the daily mean temperature.

In this study, precipitation was calculated by the sum of daily precipitation, and insolation was represented by the mean of daily downward shortwave radiation 60 days before the green-up date. Meteorological variables were calculated on the basis of gridded daily temperature, precipitation, and insolation data at the spatial resolution of 0.1° × 0.1°, which were resampled into 500-m resolution surface data by the Nearest-Neighbors method. The Nearest-Neighbors method directly took the pixel value closest to a pixel position as the new value of the pixel [58]. The advantages were that the method was simple, the processing speed was fast, and the original pixel value would not be changed. The annual AGDD requirement over 15 years (2001-2015) was used to analyze spatial changes and the relationships between AGDD and CD, AGDD and precipitation, and AGDD and insolation, respectively. For interannual variations, the slope of the linear regression between the AGDD requirement and the year was determined as the time trend of the AGDD requirement over the study period (2001-2015). A partial correlation method was used to investigate the influence of CD, precipitation, and insolation on the interannual variation of the AGDD requirements. A partial correlation analysis refers to the process that when two variables are related to the third variable at the same time, the influence of the third variable is removed, and only the correlation degree between the other two variables is analyzed, and the determination index is the value of the partial correlation coefficient. This method was successfully applied to eliminate the covariate effect between multiple influencing factors [27,34].

### 2.5. Trend Analysis Method

The Mann-Kendall [59,60] method was used to examine the trends in the green-up date and meteorological indicators. Since the Mann-Kendall method is a nonparametric test for monotonic trends, it does not assume a specific distribution for the data and is insensitive to outliers. The Mann-Kendall method was a climate diagnosis and prediction technology. It could determine whether there was a mutation in the time series, and if there was, the time of the mutation can be determined. The Theil-Sen method was a nonparametric statistical method for the significance test of the trend [61]. It was a method for robust linear regression that chooses the median slope among all lines through pairs of two-dimensional sample points. Combining the two is an excellent method of the time series trend analysis, which has been widely used in climate and hydrological trend research in recent years [62,63].

## 2.6. Flow Chart

Flow chart of the key indicators computation and statistical analysis is displayed in Figure 5. Green-up date was extracted based on the NDPI calculated from remote sensing data (MOD09A1). Meteorological indicators were calculated from the temperature, precipitation, and insolation according to the green-up date. A partial correlation analysis was used to investigate the relationship between the heat requirement indicator and meteorological indicators.

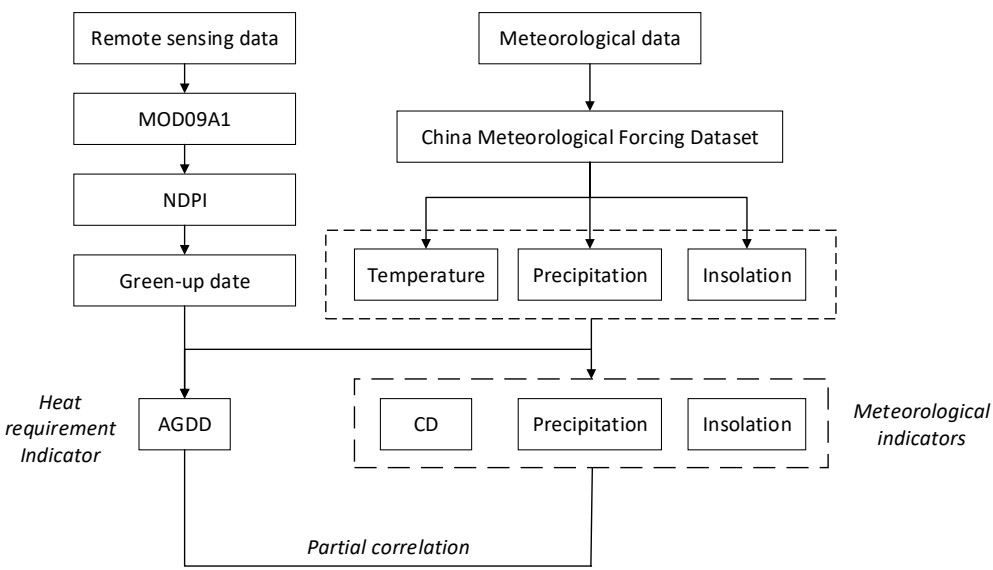

**Figure 5.** Flow chart of the key indicator computations and statistical analysis. NDPI: normalized difference phenology index. AGDD: accumulated growing degree-days. CD: chilling days.

## 3. Results

### 3.1. Spatial and Temporal Patterns of Remotely Sensed Green-Up Dates

The multiyear mean remote sensing green-up dates (Figure 6) ranged from DOY (day of year) 104 in warm and dry areas to DOY 144 in cold and wet areas across the Hulun Buir grassland; the analysis also revealed spatial variations that were delayed from the west and east to the central region.

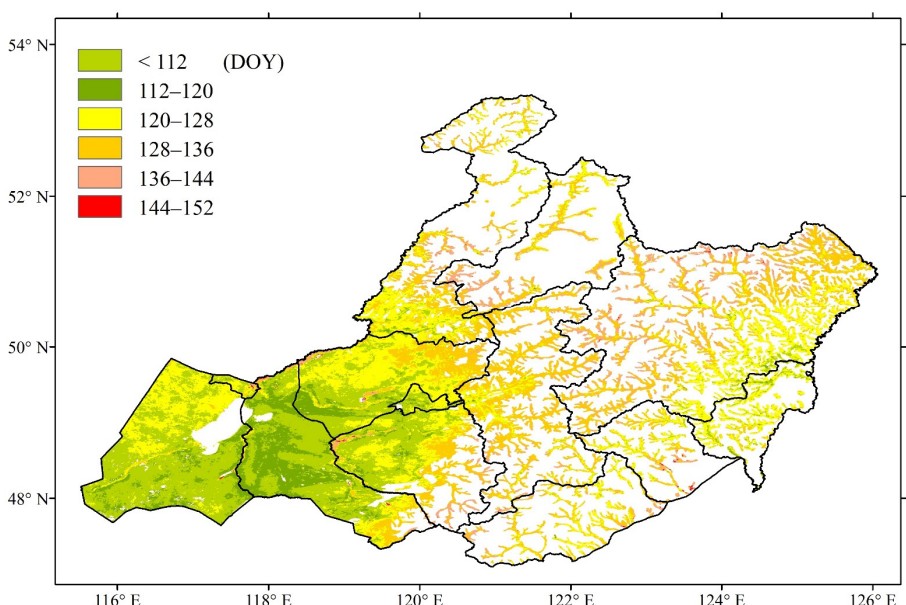

**Figure 6.** Mean green-up dates between 2001 and 2015.

Figure 7 illustrates the spatial pattern of the green-up date standard deviations, which decreased from low-attitude areas to high-attitude areas. The mean standard deviation across the whole area was 10.6 days. The statistical analysis of the number of pixels in main grassland types is shown in Figure 8. The temperate steppe had the earliest green-up date (DOY 115 ± 11.6; mean ± standard deviation), followed by the temperate meadow steppe (DOY 123 ± 8.6), lowland meadow (DOY 126 ± 10.5), and upland meadow (DOY 127 ± 7.3). The green-up date across 79.5% of the temperature steppe ranged from DOY 104 to DOY 128, 80.2% of the temperature meadow steppe ranged from DOY 112 to DOY 128, 70.1% of the lowland meadow ranged from DOY 112 to DOY 136, and 92.7% of the upland meadow ranged from DOY 120 to DOY 136.

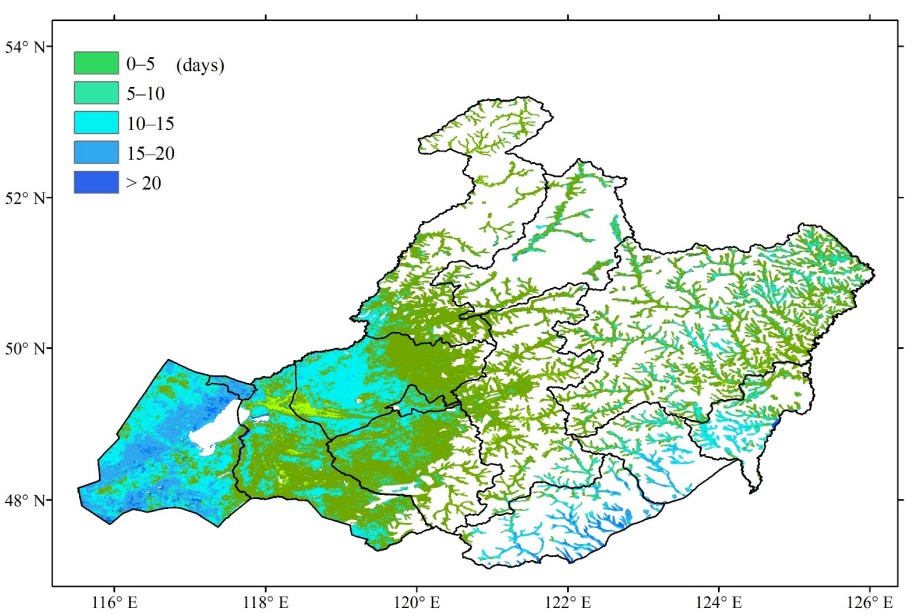

**Figure 7.** Standard deviations of the green-up dates.

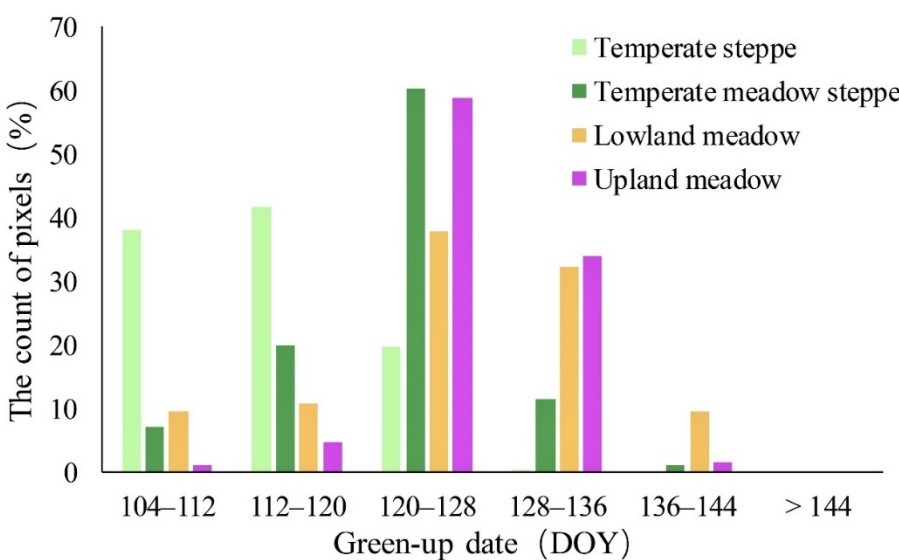

**Figure 8.** Green-up dates in the different types of grasslands.

The trends in green-up dates decreased from the west and east to the central region (Figure 9) and ranged from -15 days per decade to 15 days per decade. The mean trend

overall was -2.0 days per decade. However, only 4.4% of the pixels exhibited significant ($p < 0.05$) positive changes where mainly distributed in the northwestern temperate steppe; 9.3% of pixels exhibited significant ($p < 0.05$) negative changes where mainly distributed in the temperate meadow steppe and northeast lowland meadow. The areas with large delayed trends were primarily concentrated in the temperate steppe, whereas the areas with large advanced trends were primarily concentrated in the lowland meadow areas.

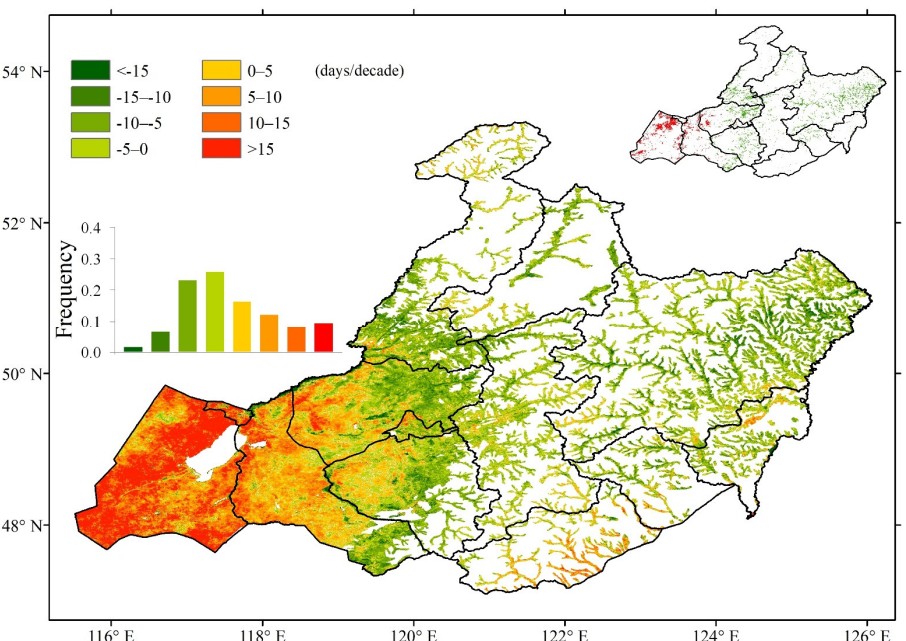

**Figure 9.** Spatial pattern of the temporal trends in green-up dates (in days per decade) between 2001 and 2015. The inset shown at the top-right of the figure indicates pixels with a significant ($p < 0.05$) increase (red) or decrease (green). The middle-left inset shows the frequency distribution of trends corresponding to the values indicated by the map legend.

### 3.2. Trend Analysis for AGDD Requirement

During the study period, 71.3% of the study area showed an increasing trend in AGDD requirement (Figure 10). Pixels showing significant changes in AGDD requirement ($p < 0.05$) accounted for 9.3% of the entire study area. Of these pixels, 7.8% of the study area showed a significant increase; these were primarily distributed in the temperate steppe area of New Barag Right Banner.

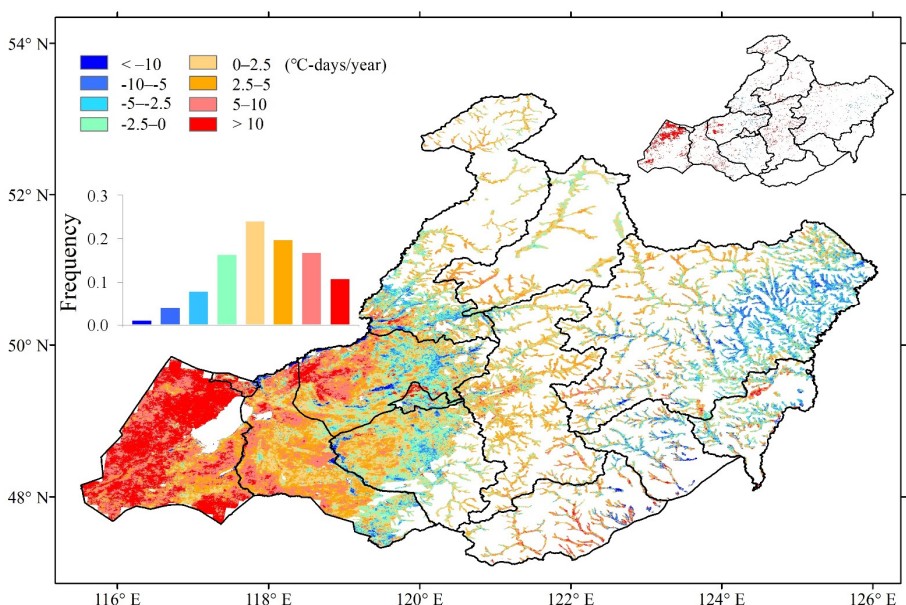

**Figure 10.** Spatial pattern of the temporal trends in accumulated growing degree-days requirements (in °C-days year$^{-1}$) between 2001 and 2015. The top-right inset indicates pixels with a significant ($p < 0.05$) increase (red) or decrease (blue). The middle-left inset shows the frequency distribution of trends corresponding to the values indicated by the map legend.

### 3.3. Trend Analysis of Meteorological Indicators

Meteorological indicators showed different trends in the Hulun Buir grassland area over the preseason. The trend for CD (Figure 11a) in 75.6% pixels ranged from -0.6 days year$^{-1}$ to 0.6 days year$^{-1}$. Only less than 1% area in the CD trend was statistically significant ($p < 0.05$).

The trend for precipitation (Figure 11b) in 93.6% of the pixels ranged from -3 to 1 mm year$^{-1}$. Of these pixels, 16.4% showed a significant trend for reduced precipitation ($p < 0.05$), of which 74% were distributed in the eastern temperate steppe and northeastern lowland meadow.

The trend for insolation over the preseason (Figure 11c) was approximately -2 to 3 W m$^{-2}$ year$^{-1}$. There was a significant ($p < 0.05$) increasing trend for insolation over 20.3% of the region, primarily concentrated in the temperate steppe. Less than 1% of the pixels showing a decreasing trend for insolation were statistically significant ($p < 0.05$).

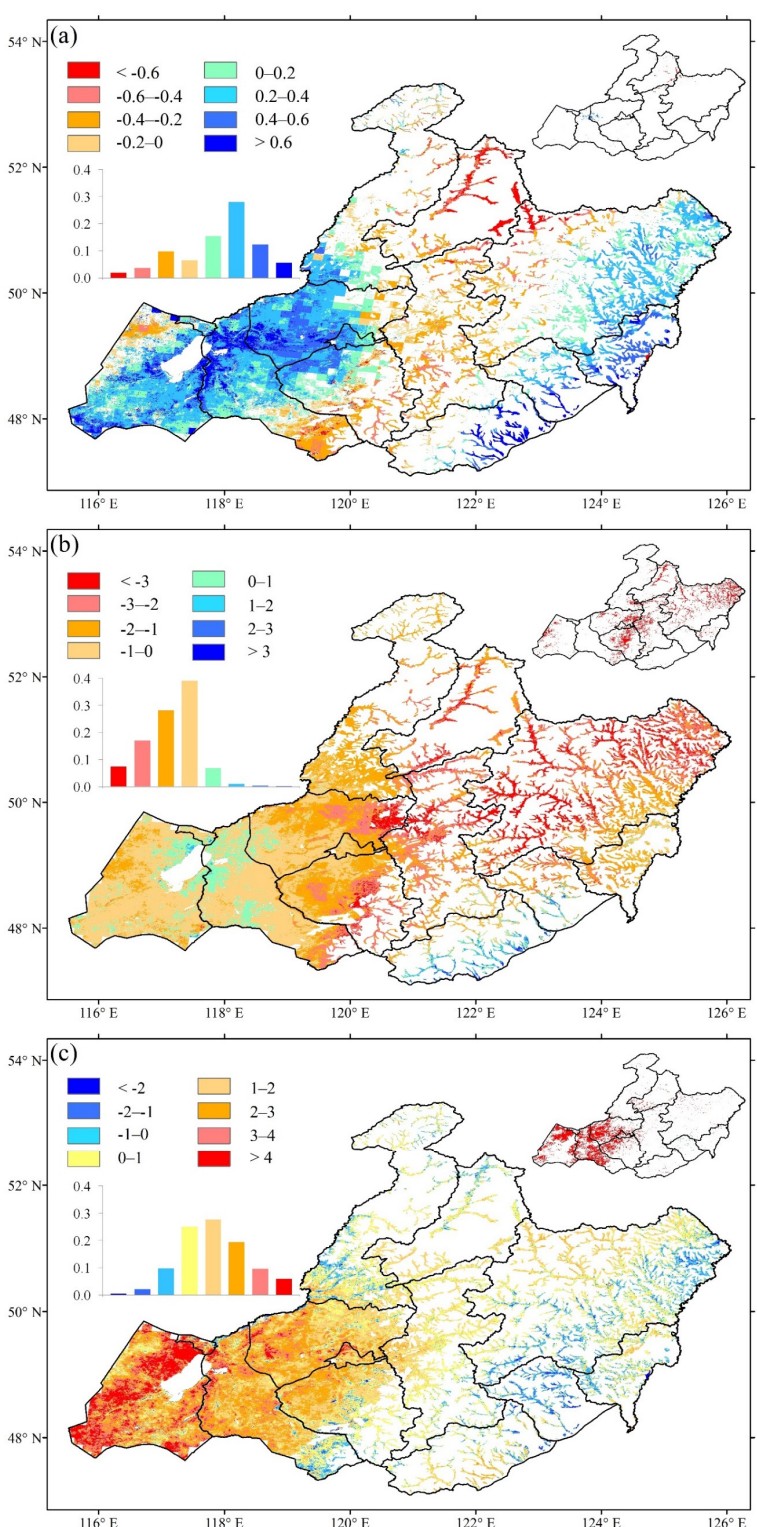

**Figure 11.** Spatial pattern of the temporal trends in (a) chilling days in days year⁻¹, (b) precipitation in mm year⁻¹, and (c) insolation in W m⁻² year⁻¹. The top-right insets indicate pixels with a significant (*p* < 0.05) increase (blue in (a) and (b) and red in (c)) or decrease (red in (a) and (b) and blue in (c)). The middle-left insets show the frequency distributions of trends corresponding to the values indicated by the map legends.

### 3.4. Partial Correlation Analysis between AGDD Requirement and Environmental Factors

3.4.1. AGDD Requirement and CD

Across the entire region, 99.4% of pixels showed a negative partial correlation between the AGDD requirement and CD (Figure 12a), of which 67.0% of the pixel partial correlation coefficients were less than -0.55 (0.55 corresponds to $p = 0.05$). Furthermore, 38.3 % of the pixel partial correlation coefficients were less than −0.68 (0.68 corresponds to $p = 0.01$).

3.4.2. AGDD Requirement and Precipitation

Across the entire region, 80.4% of the pixels showed a positive partial correlation between the AGDD requirement and precipitation (Figure 12b), of which 28.9% of the pixel partial correlation coefficients were greater than 0.55. Seventeen point seven percent of the pixel partial correlation coefficients were greater than 0.68. No pixels showed a significant ($p < 0.05$) negative partial correlation.

3.4.3. AGDD Requirement and Insolation

Compared with CD and precipitation, insolation had greater influence on the AGDD requirements (Figure 12c). Across the entire region, 97.6% of the pixel partial correlation coefficients were greater than 0.55. Furthermore, 93.3% of the pixel partial correlation coefficients were greater than 0.68.

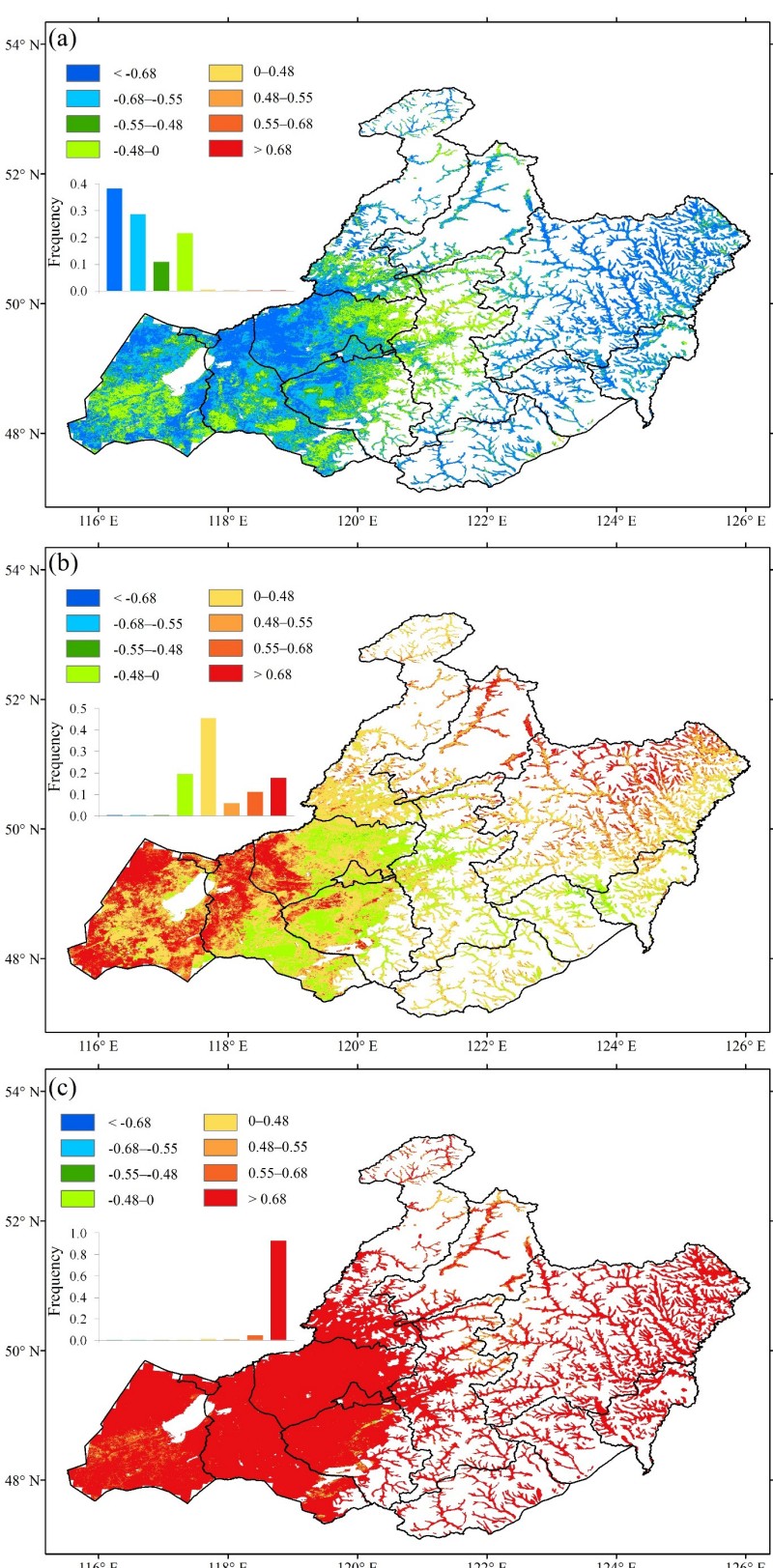

**Figure 12.** Spatial patterns of the interannual partial correlations between the accumulated growing degree-days requirement and chilling days (a), precipitation (b), and insolation (c). Partial correlation coefficient values of ± 0.68, ± 0.55, and ± 0.48 correspond to significance at *p* = 0.01, *p* = 0.05, and *p* = 0.10, respectively. Top-left insets show the frequency distributions of the correlation coefficients corresponding to values indicated by the map legends.

*3.5. Partial Correlation Analysis between AGDD Requirement and Environmental Factors in Different Grassland Types*

3.5.1. Temperate Steppe

Overall, 81.3% of pixels showed a significant ($p < 0.05$) negative correlation between AGDD requirement and CD. Furthermore, 21.5% of pixels showed a significant ($p < 0.05$) positive correlation between AGDD requirement and precipitation; these were primarily distributed in New Barag Right Banner. Ninety-seven percent of pixels showed a significant ($p < 0.05$) positive correlation between AGDD requirement and insolation.

3.5.2. Temperate Meadow Steppe

Forty-three point five percent of the pixels showed a significant ($p < 0.05$) negative correlation between the AGDD requirement and CD and were primarily distributed in Chen Barag Banner. Eight point seven percent of the pixels showed a significant ($p < 0.05$) positive correlation between the AGDD requirement and precipitation. Seventy-two point three percent of the pixels showed a significant ($p < 0.05$) positive correlation between the AGDD requirement and insolation.

3.5.3. Lowland meadow

Fifty-three point seven percent of pixels showed a significant ($p < 0.05$) negative correlation between the AGDD requirement and CD and were primarily distributed in other than the middle area. Fifteen percent of pixels showed a significant ($p < 0.05$) positive correlation between the AGDD requirement and precipitation and were primarily distributed in the southwestern and northeastern regions. Furthermore, 73.7% of pixels showed a significant ($p < 0.05$) positive correlation between the AGDD requirement and insolation and were primarily distributed in other places than the high-altitude region.

3.5.4. Upland Meadow

A significant negative correlation ($p < 0.05$) was detected between the AGDD requirement and CD for 21.2% of pixels. A significant positive correlation ($p < 0.05$) was detected between the AGDD requirement and precipitation for 2.4% of pixel. Finally, 58% of pixels showed a significant ($p < 0.05$) positive correlation between the AGDD requirement and insolation.

## 4. Discussion

*4.1. Factors Affecting the Spatial Variation of AGDD Requirement*

This study revealed spatial variations in AGDD requirement throughout the Hulun Buir grassland vegetation. This variability indicates the need to use AGDD requirements at the pixel scale in order to establish a model of vegetation phenology. On the east and west sides of the Greater Khingan Mountains, the grassland was primarily lowland meadows, and the AGDD requirement was relatively low in cold regions; these findings were consistent with those published previously [28]. Some researchers consider this relationship to be the result of plants adapting to colder growing environments [36] or plants using heat more efficiently in cooler areas [28]. Plants in relatively warm regions could accumulate more heat quickly than cold regions to trigger green-up. Though a higher AGDD was required in warm steppe, the green-up date was earlier than other cold grassland-type areas. However, the annual mean temperature in these regions was 0.25°C higher than that of the lowland meadow area, and the AGDD requirement was 11°C-days higher than that of the lowland meadow area. The temperate steppe area showed the earliest green-up date for plants across the entire study area with the least precipitation and relatively high temperatures. Due to favorable thermal conditions, this relatively dry area may exhibit high levels of efficiency in terms of water use, thus allowing vegetation to break their dormant period earlier and enter a period of ecological growth.

### 4.2. Changes in AGDD Requirement

Previous analysis showed that the speed of climate warming in most parts of the world weakened or stagnated during the period between 2002 and 2012 [64]. However, between 2001 and 2015, the Hulun Buir grassland area showed no significant change in terms of the annual mean temperature. Previous studies reported that the warming trend in Hulun Buir between 2001 and 2013 was weaker than that between 1970 and 2000 and that the largest reduction occurred in the winter warming trend; in contrast, the largest increase over an entire year was for the spring season warming [45]. However, the AGDD requirement increased significantly in only 18.4% of the region, as reported previously for more than 10 temperate trees in Europe and North America [27]. Our analysis showed that 90.2% of the regional AGDD requirement did not change significantly; this proportion was the same as that described previously for the change in AGDD requirement on the Qinghai-Tibet Plateau [34]. We also found that the increase in heat requirement for grassland vegetation was positively correlated with precipitation. In areas with an earlier green-up date, the AGDD requirement was higher than that in areas with a later green-up period because of the high temperature. This result is consistent with previous studies [27,65]. The AGDD requirement for the later green-up dates was relatively large (Figure 13), and the altitudes of these area were high. Climate warming caused warm weather, and more chilling days were required to break the dormant period [66].

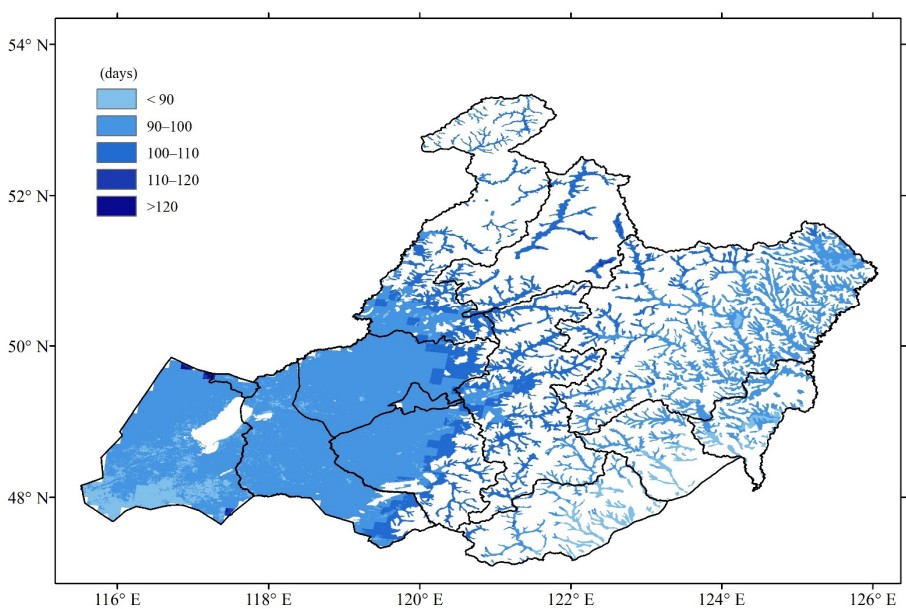

**Figure 13.** The average annual chilling days (CD) during 2001-2015.

### 4.3. Factors Affecting Interannual AGDD Requirement

The accumulation of heat is recognized as a major factor in triggering the green-up of plants [37,40]. However, little is known of the mechanisms responsible for the relationship between heat requirement and meteorological indicators over the preseason. For the Tibetan Plateau vegetation and for European and North American trees, the heat requirement and CD accumulation are primarily negatively correlated [27,34]. Our present findings are in consistent with those described by previous research in that heat requirement and CD accumulation were negatively correlated. The dependence of AGDD on CD has been recognized previously but is not yet fully understood [44]. The negative partial correlation between AGDD requirement and CD means that if the CD decease, the winter warming continues, the number of chilling days will decrease in the future, and the AGDD requirements will increase. Plants would break dormancy when chilling requirements were fulfilled. Currently, it is difficult to determine the minimum number of CD

required [44]. Only comparisons between different regions and different vegetation types in climate change control experiments will help us to understand the specific CD required for vegetation green-up.

Our studies further showed that in 28.9% of the Hulun Buir grassland, there was significant ($p < 0.05$) positive correlation between the AGDD requirement and precipitation over the preseason. In nearly 70% of the Hulun Buir grassland, there was no significant correlation between the AGDD requirement and precipitation. This indicates that increased precipitation did not significantly reduce the heat requirement. Furthermore, the effect of snowmelt on phenology in spring was confirmed by a snow cover experiment [67]. A positive correlation has also been reported between the heat requirement and precipitation [27]. In the west warm steppe, the precipitation in the winter was equal with April. An increase in insolation can reduce the water content of soil by increasing surface evaporation, thus delaying the green-up date and increasing AGDD requirement.

Significant correlation was detected between AGDD requirement and insolation in most of our study regions, and an increase in insolation led to a significant increase in AGDD requirements. Previous studies in European woody species showed that total insolation could directly adjust the AGDD requirement and that this occurred independent of any chilling effect [68]. These previous studies were based on woody plants, and the response of grassland vegetation to total insolation remains poorly understood. The response of different species of plants to total insolation must also be considered. Approximately 95% of our study regions showed a significant ($p < 0.05$) positive correlation with insolation. Relatively high insolation may result in relatively high temperatures. In this area, snowfall in the winter was equal with the total precipitation in April. An increase in insolation can reduce the water content of soil by increasing surface evaporation, thus delaying the green-up date and increasing the AGDD requirement.

A previous study reported that temperatures slightly above freezing were the most effective in satisfying the chilling requirement [69] and suggested that the temperature range between 0 and 5 °C is the most effective across species. In our study area, the most effective temperature was unclear. At the same time, the most effective time range in precipitation and insolation were also unclear. So, we investigated the partial correlation under a 0°C threshold 30 days before the green-up date (Figure 14) and a 5°C threshold 60 days before the green-up date (Figure 15). By contrast, the correlations over the whole area were similar with the correlations under the 0°C threshold 60 days before the green-up date, although the level was weakened. The 0°C threshold could be better to explain the partial correlations between the AGDD requirement and meteorological indicators 60 days before the green-up date.

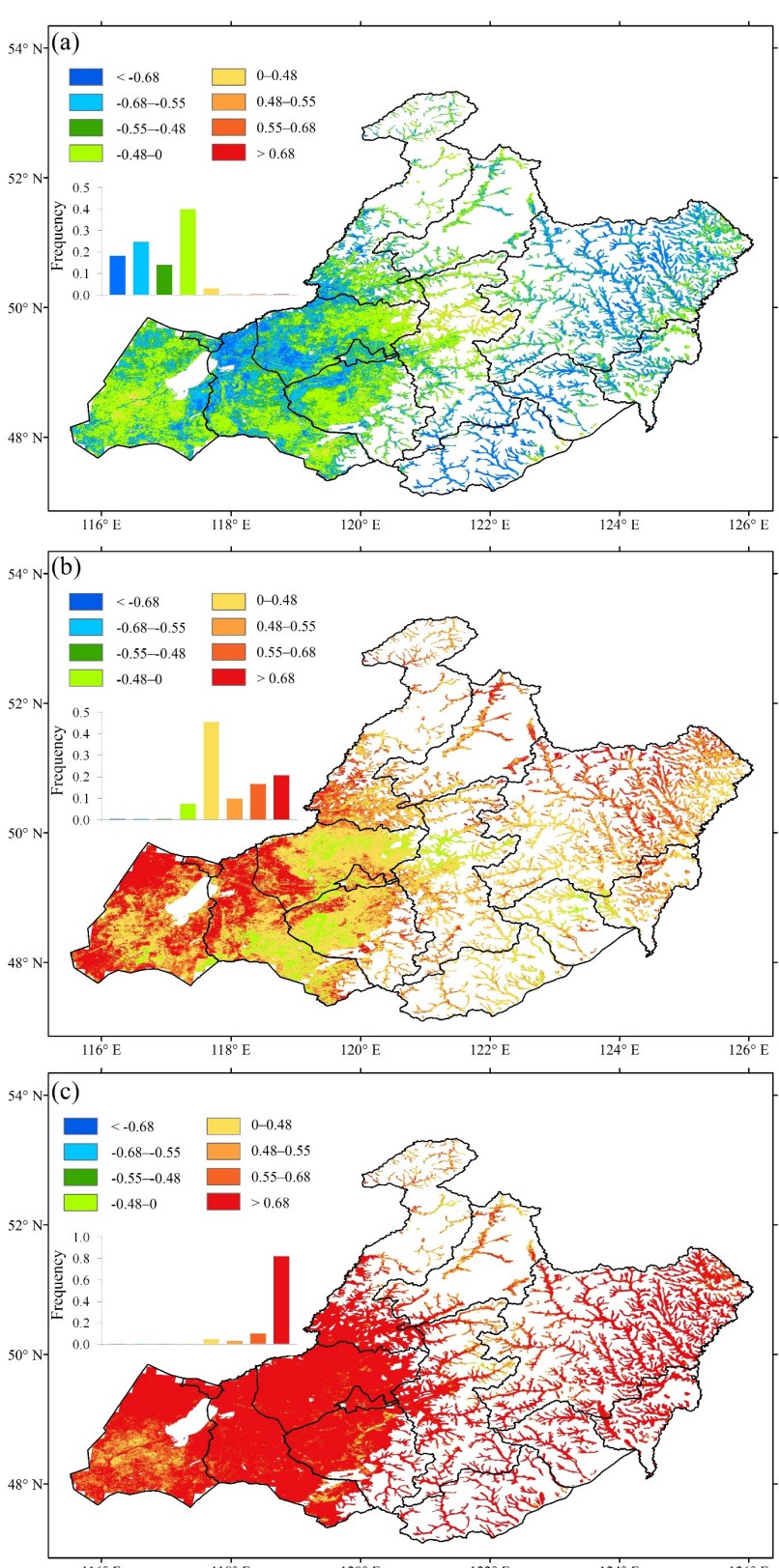

**Figure 14.** Spatial patterns of the interannual partial correlations between the accumulated grow-ing degree-days requirements and chilling days (a), precipitation (b), and insolation (c) under the 0°C threshold 30 days before the green-up date. Partial correlation coefficient values of ± 0.68, ± 0.55, and ± 0.48 correspond to significance at *p* = 0.01, *p* = 0.05, and *p* = 0.10, respectively. Top-left insets show the frequency distributions of the correlation coefficients corresponding to the values indicated by the map legends.

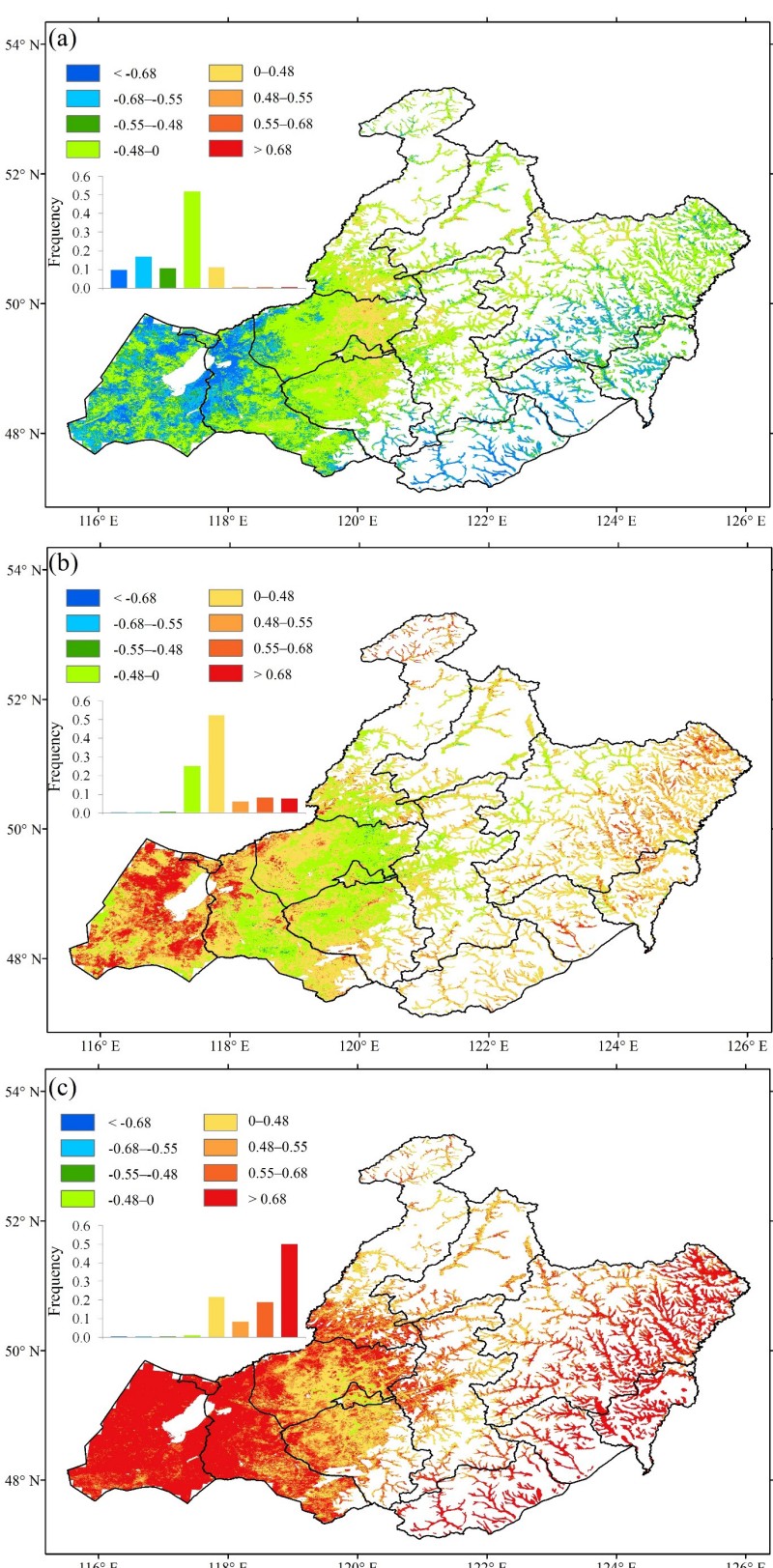

**Figure 15.** Spatial patterns of the interannual partial correlations between the accumulated grow­ing degree-days requirement and chilling days (a), precipitation (b), and insolation (c) under the 5°C threshold 60 days before the green-up date. Partial correlation coefficient values of ± 0.68, ± 0.55, and ± 0.48 correspond to significance at *p* = 0.01, *p* = 0.05, and *p* = 0.10, respectively. Top-left insets show the frequency distributions of the correlation coefficients corresponding to the values indicated by the map legends.



## 5. Conclusions

In the present study, remote sensing and meteorological data were used to investigate the temporal and spatial variations of long-term green-up dates and AGDD requirements in the Hulun Buir grasslands during the winter and spring warming periods between 2001 and 2015. The green-up date was significantly different for different grassland types and showed large spatial variations ranging from DOY 104 in warm and dry areas to DOY 144 in cold and wet areas. Overall, there was an advancing trend for green-up dates of -2.0 days per decade. The AGDD requirements trend did not significantly change in most of the area. We conclude that the interannual variations in the AGDD requirements were extensively driven by the number of chilling days and mean of the insolation, while precipitation affected the AGDD requirements in limited areas. The chilling days and isolation have more impact on the heat requirements for the vegetation green-up than precipitation in the context of climate change.

**Author Contributions:** Conceptualization, J.G. and X.Y.; software, J.G. and M.Z.; formal analysis, J.G., F.C., Y.J., and A.C.; data curation, J.G., G.S., X.X., and D.Y.; writing—original draft preparation, J.G.; writing—review and editing, J.G., X.Y., J.N., S.L., and B.X.; and funding acquisition, X.Y. All authors have read and agreed to the published version of the manuscript.

**Funding:** This study was supported by the National Key Research and Development Program of China (2017YFC0506504) and National Natural Science Foundation of China (41571105, 41861019, and 31372354).

**Conflicts of Interest:** The authors declare no conflicts of interest. The funders had no role in the design of the study; in the collection, analyses, or interpretation of data; in the writing of the manuscript; or in the decision to publish the results.

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
