# Peer review of "Examining Relationships between Heat Requirement of Remotely Sensed Green-Up Date and Meteorological Indicators in the Hulun Buir Grassland"

_remotesensing, doi:10.3390/rs13051044_

Round 1

Reviewer 1 Report

Comments attached

Reviewer 2 Report

This research is an informative and comprehensive investigation of phenological phenomena in a grassland of global and regional significance. Your review of the literature was well-structured and clear. Methods were described in detail (such as your reasons for using NDPI rather than NDVI) and are easy to follow. The maps are visually appealing and illustrate your results well. I especially appreciated the organization of the discussion section, which broke things down spatially and temporally and situated your results in the larger context of climate change. My suggested changes are minor. They follow below, by line:

28: suggest restructuring sentence along the lines of "9.2% of the study area had significant changes in AGDD"

32: this sentence is incoherent as written

76: should say "Consequently, it is necessary"

83: Is it spatial? Seems to describe more of a temporal phenomena

104: Would be a good place to define chilling days in more detail, as you did with other terminology like AGDD or NDVI/NDPI

238: extra space in sentence

492: should be "significant change"

495: should be "areas" plural

500: suggest adding a concluding sentence situating your results in the context of climate change, as you ably did in the discussion section
